# Tumor-Targeted Erythrocyte Membrane Nanoparticles for Theranostics of Triple-Negative Breast Cancer

**DOI:** 10.3390/pharmaceutics15020350

**Published:** 2023-01-20

**Authors:** Moon Jung Choi, Yeon Kyung Lee, Kang Chan Choi, Do Hyun Lee, Hwa Yeon Jeong, Seong Jae Kang, Min Woo Kim, Young Myoung You, Chan Su Im, Tae Sup Lee, Yong Serk Park

**Affiliations:** 1Department of Biomedical Laboratory Science, Yonsei University, Wonju 26493, Republic of Korea; 2Division of RI Application, Korea Institute of Radiological and Medical Sciences (KIRAMS), Seoul 01812, Republic of Korea

**Keywords:** erythrocyte-derived nanoparticle, drug delivery system, triple-negative breast cancer, tumor-targeted therapy, quantum dots

## Abstract

Triple-negative breast cancer (TNBC) cells do not contain various receptors for targeted treatment, a reason behind the poor prognosis of this disease. In this study, biocompatible theranostic erythrocyte-derived nanoparticles (EDNs) were developed and evaluated for effective early diagnosis and treatment of TNBC. The anti-cancer drug, doxorubicin (DOX), was encapsulated into the EDNs and diagnostic quantum dots (QDs) were incorporated into the lipid bilayers of EDNs for tumor bio-imaging. Then, anti-epidermal growth factor receptor (EGFR) antibody molecules were conjugated to the surface of EDNs for TNBC targeting (iEDNs). According to the confocal microscopic analyses and biodistribution assay, iEDNs showed a higher accumulation in EGFR-positive MDA-MB-231 cancers in vitro as well as in vivo, compared to untargeted EDNs. iEDNs containing doxorubicin (iEDNs-DOX) showed a stronger inhibition of target tumor growth than untargeted ones. The resulting anti-EGFR iEDNs exhibited strong biocompatibility, prolonged blood circulation, and efficient targeting of TNBC in mice. Therefore, iEDNs may be used as potential TNBC-targeted co-delivery systems for therapeutics and diagnostics.

## 1. Introduction

Breast cancer, the most common cancer type in the global female population, is highly heterogeneous and complex in terms of biological characteristics. Breast cancers are classified into four subgroups depending on the cell surface receptors such as estrogen receptor (ER), progesterone receptor (PR), and human epidermal growth factor receptor 2 (HER2): luminal B (ER+, PR+, HER2+), luminal A (ER+, PR+, HER2−), HER2 overexpression (ER−, PR−, HER2+), and triple-negative breast cancer (TNBC; ER−, PR−, HER2−) [1]. These receptors are often targeted for immunotherapy of breast cancer [2]. Among these subtypes, TNBC is considered to be the most malignant and accounts for 15% of all breast cancer cases. TNBC does not express these receptors, which leads to poor prognosis and difficulties in its diagnosis and treatment. However, TNBC has been found to highly express the epidermal growth factor receptor (EGFR) [3]; therefore, this receptor has been targeted for treatment.

A great deal of effort has been expended to develop methods to enhance the therapeutic efficacy of anti-cancer drugs while reducing their aberrant side effects. Some nanocarriers were able to partially fulfill these research goals [4,5]. Recently, many research groups have attempted to incorporate membrane grafting in their nanocarrier studies [6,7]. Various types of camouflaged particles made of polymers and metal particles have been coated with cell membranes on their surfaces to improve their circulation in blood [8]. In fact, particles coated with cell membranes circulate in the blood longer than the bare particles [9]. In particular, the plasma membranes of red blood cells (RBCs) have been extensively tested as they are the most abundant cells in the blood and circulate for 120 d. The membrane structure of RBCs is flexible, and its composition is relatively simple compared to that of other cell types. In particular, CD47, present on the surface of red blood cells, is known to exhibit a ‘don’t eat me’ signal by associating with the signal-regulatory protein-alpha (SIRPα) on the surface of phagocytic cells [10,11,12]. This interaction causes RBCs to circulate in the blood for a long time without eradication by immune cells. Therefore, erythrocyte membranes are widely tested to improve the systemic circulation of nanoparticles [13,14]. However, camouflaged particles still show limitations such as the toxicity of core particles and unintended off-targeting.

Many previous groups have suggested biomembrane-derived nanoparticles as a biocompatible drug delivery system, but most of them used the biomembranes to coat or cover the artificial core molecules such as poly(lactic-co-glycolic acid) (PLGA) particles [15] or metals [16]. In this study, biocompatible erythrocyte membrane-derived nanoparticles (EDNs) were developed for tumor-targeted treatment and tumor imaging. To prepare theranostic vehicles, doxorubicin (DOX) was directly encapsulated inside EDNs, and hydrophobic quantum dots (QDs) were incorporated into the lipid bilayer of EDNs. To provide targetability to TNBCs, anti-EGFR antibodies were conjugated to the surface of EDNs. This system was expected to reduce the side effects of DOX [17] and provide in vivo tumor images during treatment. This novel theranostic drug delivery system targeting TNBCs was also systemically evaluated in in vitro and in vivo tumor models.

## 2. Materials and Methods

### 2.1. Materials

The chemicals, 1,2-distearoyl-sn-glycero-3-phosphoethanolamine-N-[18] (DSPE-PEG2000) and 1,2-distearoyl-sn-glycero-3-phosphoethanolamine-N-[maleimide(polyethylene glycol)2000] (DSPE-PEG2000-mal), were purchased from Avanti Polar Lipid, Inc. (Alabaster, AL, USA). Anti-EGFR antibody (cetuximab, Erbitux^®^) was purchased from Merck (Darmstadt, Germany). QDs were purchased from ZEUS (Hwaseong, Republic of Korea). The PD-10 column and sepharose CL-4B were purchased from Amersham Bioscience (Uppsala, Sweden). Ultra-4 30 K MWCO was purchased from Amicon (Millipore, Darmstadt, Germany). Cell Counting kit-8 (CCK-8) was supplied by Dojindo Laboratories (Kumamoto, Japan). Doxorubicin hydrochloride was purchased from Merck (Darmstadt, Germany). Vyrant DiO Cell-Labeling Solution was purchased from Thermo Fisher Scientific (Waltham, MA, USA).

### 2.2. Cell Lines and Cell Culture

Human breast adenocarcinoma cells [MDA-MB-231 (HTB-26™) and MDA-MB-453 (HTB-131™)] and mouse macrophages (RAW 264.7) were purchased from the American Type Culture Collection (ATCC, Manassas, VA, USA). MDA-MB-231 and MDA-MB-453 cells were maintained in Leibovitz’s L-15 medium (WELGENE, Gyeongsan, Korea) supplemented with 10% fetal bovine serum (FBS, WELGENE), 100 IU/mL penicillin, and 100 μg/mL streptomycin (Gibco, Waltham, MA, USA) in a humidified and carbon dioxide (CO2)-free atmosphere at 37 °C. RAW 264.7 cells were maintained in Dulbecco’s modified Eagle’s medium (DMEM; WELGENE) supplemented with 10% FBS (WELGENE), 100 IU/mL penicillin, and 100 μg/mL streptomycin (Gibco) in a humidified atmosphere of 95% air and 5% CO_2_ at 37 °C.

### 2.3. Animals

Six-week-old female BALB/c nude mice and 8-week-old female ICR mice were purchased from Nara Biotech (Seoul, Korea). All animal experiments were approved by the Institutional Animal Care and Use Committee (IACUC) of Yonsei University, Mirae Campus (YWCI-201706-011-03) and performed in accordance with the school guidelines and regulations.

### 2.4. Preparation of Immuno-EDNs (iEDNs)

Mouse whole blood (~1 mL) was collected from an ICR mouse with microtainer tubes containing K2EDTA (Becton, Dickinson and Company, NJ, USA) and centrifuged at 500× *g* for 15 min at 4 °C. The serum and buffy coat were removed, and the RBCs were washed twice with 1 mL phosphate-buffered saline (PBS) by centrifugation, as described above. The washed RBCs were suspended in 1 mL of hypotonic PBS (0.25×) and centrifuged at 21,000× *g* for 15 min at 4 °C. The precipitated erythrocyte ghost membranes (1 mg of membrane proteins) were collected and sonicated three times for 15 min at 10 min intervals using a bath sonicator (Elma Schmid Bauer GmbH, Singen, Germany) at 37 kHz/60 W.

To prepare EDNs containing QDs, DSPE-mPEG2000 and QDs were mixed in chloroform/methanol (CHCl_3_/MeOH; 2:1) at a molar ratio of 350:1 according to a previous report [19]. The mixture of lipid and QDs was evaporated under a stream of N2 gas and thoroughly dried by vacuum desiccation for 1 h. The lipid film was hydrated with 4-(2-hydroxyethyl)-1-piperazineethanesulfonic acid (HEPES) buffer (HEPES 25 mM, NaCl 140 mM, pH 7.4, 1 mg lipid/mL). Uncaptured QD aggregates were removed by filtering through a 100 nm polycarbonate membrane filter (Whatman Inc., Piscataway, NJ, USA). Then, the prepared QD-micelles were mixed with 1 mg of erythrocyte ghosts and sonicated at 4 °C until they became transparent. The prepared EDNs were quantified by the Bradford assay (Bio-Rad, Hercules, CA, USA) for membrane proteins and the phosphate assay for phospholipids [20]. The amount of EDNs utilized in the following experiments is equal to the amount of quantified membrane proteins in EDNs.

To prepare the micelle solution of DSPE-PEG2000-mal, the dried lipid (1 mg) was hydrated with 2 mL of HEPES-EDTA buffer. The solution of DSPE-PEG2000-mal (100 μg) was added to the EDN solutions (1 mg) and sonicated three times for 15 min with 10 min intervals on ice. In another tube, 300 μg of cetuximab (19.7 nmol) was thiolated by incubation with 2 μg of Traut’s reagent (Thermo Scientific, Rockford, IL, USA) for 1 h at room temperature. Freshly thiolated antibodies were immediately added to the EDNs exposed to maleimides (3.94 nmol) and then incubated overnight at 4 °C. Cetuximab and EDN conjugates were purified by gel filtration chromatography on Sepharose CL-4B columns in HEPES buffer (pH 7.4). Then 10 μL of all fractions were added to 96 wells with 200 μL of Bradford reagent, and the absorbance of protein-dye complex in each well was measured at 595 nm [21]. Antibody conjugation to EDNs was determined by 6% sodium dodecyl sulfate-polyacrylamide gel electrophoresis (SDS-PAGE). Antibody coupling efficiency was estimated by measuring the density of the stained bands using Fusion Solo Chemidoc (Vilber Lourmat, Eberhardzell, Germany).

### 2.5. DOX Encapsulation into iEDNs

DOX was encapsulated into the iEDNs using the phosphate-gradient method [20]. Briefly, iEDNs (1 mg) were prepared in 300 mM ammonium phosphate buffer (pH 7.4) and then passed through a PD-10 column equilibrated with the HEPES buffer (25 mM HEPES, pH 7.4). An aliquot of DOX solution (5 mg/mL; Sigma Aldrich, St. Louis, MI, USA) was added to the prepared iEDNs (1:8 weight ratio of DOX and EDN) and incubated for 48 h at 37 °C. Free DOX was removed by gel filtration using a PD-10 column, equilibrated with HEPES buffer, and the fractions of iEDNs-DOX were concentrated by centrifugation in a 30 K Amicon tube (Millipore) for 10 min at 3500× *g*. The efficiency of DOX encapsulation was analyzed by measuring the absorbance at 480 nm using an Infinite 200 Pro NanoQuant (TECAN Group Ltd., Mannedorf, Switzerland).

The cumulative release of DOX from EDNs-DOX and iEDNs-DOX was evaluated by the dialysis method using a 3.5 kDa DiaEasy Dialyzer (Biovision Inc., Mipitas, CA, USA). Free DOX, EDNs-DOX, and iEDNs-DOX (100 μg of DOX or equivalent) were added to the dialysis tubes and dialyzed in 1 L of HEPES buffer (25 mM, pH 7.4) with constant stirring for 48 h. The amount of remaining DOX was measured using an Infinite 200 Pro NanoQuant (TECAN) at a wavelength of 480 nm.

### 2.6. Characterization of iEDNs-DOX

The sizes and surface charges of the EDNs, iEDNs, EDNs-DOX, and iEDNs-DOX were measured thrice by dynamic light scattering (DLS) using Zetasizer Nano-ZS90 (Malvern Instrument Ltd., Malvern, UK). The measurements were statistically analyzed using the GraphPad Prism software 8.2.1 (GraphPad Software, Inc., San Diego, CA, USA).

The morphologies of EDNs, iEDNs, EDNs-DOX, and iEDNs-DOX were observed by transmission electron microscopy (TEM) (JEM-2100F; JEOL Ltd., Tokyo, Japan). The solutions containing the particles (3 μg of protein) were loaded onto carbonyl-coated 400 mesh copper grids. For negative staining, 10 μL of 2% uranyl acetate was placed on the grid for 10 min; it was then removed and dried for 10 min at room temperature. TEM images were acquired at an acceleration voltage of 80 kV.

### 2.7. Western Blotting Analysis

The expression levels of CD47 in RBC ghosts and EGFRs were detected by Western blotting. RBCs, MDA-MB-231 cells, and MDA-MB-453 cells (2 × 10^5^ cells per well) were lysed using 200 μL of radioimmunoprecipitation assay buffer (Thermo Scientific, Waltham, MA, USA) with a protease inhibitor cocktail (Thermo Scientific). The cell lysates were quantified by Bradford assay, run on 10% SDS-PAGE gels, and transferred to nitrocellulose membranes (Pall Corp., Port Washington, NY, USA). The membranes were blocked with Tris-buffered saline with 0.1% Tween 20 (TBST) containing 3% skim milk (BD, Franklin Lakes, NJ, USA) for 1 h at room temperature. The membranes were incubated with a 1:1000 dilution of anti-mouse CD47 antibodies (Invitrogen, Carlsbad, CA, USA), anti-EGFR rabbit antibodies (Cell Signaling Technology, Danvers, MA, USA), or 1:3000 dilution of anti-glyceraldehyde-3-phosphate dehydrogenase (anti-GAPDH) mouse antibodies (Millipore, Darmstadt, Germany) in TBST containing 3% skim milk at 4 °C overnight. After washing thrice with TBST, the membranes were treated with 1:5000 dilution of horseradish peroxidase-conjugated goat anti-rabbit antibodies or goat anti-rat antibodies (Bethyl Laboratories, Montgomery, AL, USA) or 1:3000 dilution of horseradish peroxidase-conjugated goat anti-mouse antibodies (Jackson ImmunoResearch, West Grove, PA, USA) in TBST containing 3% skim milk for 1 h at room temperature. The membranes were washed thrice with TBST and treated with SuperSignal West Pico (Thermo Scientific). The immunoreactive bands were visualized using a Fusion Solo Chemidoc (Vilber Lourmat).

### 2.8. Cytotoxicity Assay

To determine the cell toxicity of EDN formulations, MDA-MB-231, and MDA-MB-453 cells were seeded into 96-well plates (1 × 10^4^ cells/well in 200 μL of 10% FBS-containing L-15 media per well) for 24 h. The cells were treated with various concentrations of EDNs (0, 50, 100, 250, and 500 μg/mL) in 100 μL of serum-free culture media and then incubated at 37 °C for 24 h.

To verify the cytotoxicity of free DOX, EDNs-DOX, and iEDNs-DOX, the MDA-MB-231 and MDA-MB-453 cells were seeded as described above. The cancer cells were treated with various concentrations of free DOX, EDNs-DOX, and iEDNs-DOX (100, 10, 1, 0.1, 0.01, 0.001, and 0.0001 μM DOX) in 100 μL of serum-free culture medium and incubated at 37 °C for 48 h. After incubation, all the cells were washed once with the culture media, treated with 10 μL of CCK-8 solution, and incubated for 4 h at 37 °C. The wells were analyzed at 450 nm using Infinite 200 Pro NanoQuant (TECAN). Half maximal inhibitory concentration (IC_50_) values of free DOX, EDNs-DOX, and iEDNs-DOX were estimated using the GraphPad Prism software (GraphPad Software, Inc., San Diego, CA, USA).

### 2.9. In Vitro EDN Uptake by Macrophages

To evaluate the immune escapability of EDNs, their uptake by RAW 264.7 macrophages was analyzed by flow cytometry and microscopy. RAW 264.7 cells were seeded in 6-well plates at a density of 2 × 10^5^ cells/well and cultured for 12 h. Freshly prepared EDNs, CD47-blocked EDNs treated with anti-CD47 antibody (1:1000 dilution) for 1 h at 37 °C, trypsinized EDNs treated with trypsin-EDTA (0.25×), and cationic lipid nanoparticles consisting of O,O′-dimyristyl-*N*-lysyl glutamate and cholesterol (DMKE/Chol) (each 1 mg of EDNs or 1 mg of lipid nanoparticles) were fluorescently labeled with 5 μL of Vybrant Multicolor Cell-Labeling Kit (Invitrogen, Eugene, OR, USA) for 15 min at 37 °C, according to the manufacturer’s protocol. The cultured macrophages were treated with 100 μg of fluorescent EDNs, CD47 blocked-EDNs, trypsinized EDNs, and 10 μg of DMKE/Chol for 1 h at 4 °C and analyzed using a FACSCalibur flow cytometer (Becton Dickinson, Franklin Lakes, NJ, USA). The macrophages were also treated for 1 h at 37 °C and observed using a confocal laser scanning microscope (LSM 510; Zeiss, Heidenheim, Germany).

### 2.10. In Vitro Target Cell Binding and Uptake of iEDNs-DOX

To verify the target-specific binding potential of anti-EGFR iEDNs, 100 μg of EDNs or iEDNs was used to treat MDA-MB-231 (EGFR+) and MDA-MB-453 (EGFR−) cells in 6-well plates, which were further cultured for 24 h. The treated cells were washed twice with L-15, harvested with trypsin-EDTA, and centrifuged at 200× *g* for 3 min. The cell pellets were resuspended with the culture media in 5 mL polystyrene round-bottom tubes (Corning, New York, NY, USA). Alexa Fluor 488 antibody (Invitrogen, Eugene, OR, USA) was used to label the anti-EGFR antibodies for 10 min at 4 °C with continuous agitation. The binding of EDNs and iEDNs to the cells was analyzed using a FACSCalibur flow cytometer (Becton Dickinson).

To verify the cellular uptake of iEDNs-DOX, the same cancer cells were treated with 100 μg of iEDNs-DOX and incubated for 1, 4, 8, 12, or 24 h at 37 °C in a serum-free medium. After treatment, all the cells were washed twice with PBS (pH 7.4, ice-cold) and fixed with 2% paraformaldehyde at 4 °C for 10 min in the dark. After washing thrice with PBS, the cells were stained with 4’,6-diamidino-2-phenylindole solution (Vector Lab, Burlingame, CA, USA) for 10 min in the dark and mounted on slides. The slides were observed using a confocal laser scanning microscope (LSM 510; Zeiss).

### 2.11. In Vivo Biodistribution Analysis of EDNs and iEDNs

To prepare a mouse tumor model, 6-week-old female BALB/c nude mice were subcutaneously inoculated with 200 μL of MDA-MB-231 cells (1 × 10^7^) in a medium mixed with Matrigel (Corning, New York, NY, USA) (1:1 volume ratio) at the 4th mammary fat pad. Tumor volumes were measured with calipers and calculated using the formula: tumor volume = π/6 × length × width × height.

When the volume of tumors reached approximately 200 mm^3^, the mice were injected with the prepared EDNs-QD or iEDNs-QD via the tail vein (150 μg protein/mouse). Fluorescence images of the mouse whole body were analyzed with the FOBI in vivo imaging system (CELLGENTEK, Deajeon, Korea) at 4, 24, and 48 h post-injection. After taking the fluorescence images, all mice were immediately sacrificed, and major organs, including tumor tissues, were dissected. The fluorescence intensities of the organs were measured using the FOBI imaging system. To analyze EDN circulation in the blood, blood was collected from all the mice using K_2_EDTA tubes, and the absorbance of QDs was measured at 760 nm using Infinite 200 Pro NanoQuant (TECAN Group Ltd., software (Zürich, Switzerland).

### 2.12. In Vivo Tumor Growth Inhibition by iEDNs-DOX

To evaluate the anti-cancer therapeutic efficacy of iEDNs-DOX, tumor growth in mice implanted with MDA-MB-231 cells was observed for 32 d after intravenous injection of saline, free DOX, EDNs-DOX, or iEDNs-DOX thrice at 3 d intervals (n = 5, 4 mg DOX/kg). The treatment began when the average tumor volume reached approximately 100 mm^3^, and the tumor volumes and weights of the mice were measured thrice a week for 32 d. When the largest tumor volume reached approximately 1000 mm^3^, the measurements were finalized, and all mice were sacrificed.

### 2.13. Terminal Deoxynucleotidyl Transferase dUTP Nick and Labeling (TUNEL) Assay

Tumor tissues were resected after the final measurement, immediately fixed in 4% formalin, dehydrated, embedded in paraffin, and sectioned into 5 μm thickness. The presence and localization of apoptotic cells in the sections were assessed using the TUNEL assay with diaminobenzidine (DAB) staining using an in situ apoptosis detection kit (Abcam, Cambridge, UK), according to the manufacturer’s instructions. The percentage of TUNEL-positive nuclei within the total number of nuclei was calculated after the examination of the four fields for each tissue.

### 2.14. Histological Analysis

Major organs, such as the liver, heart, lungs, spleen, and kidneys, were fixed in 4% formalin for 24 h, washed with distilled water, and dehydrated using ethanol from 70% to 100%. The fixed tissues were embedded in paraffin and sectioned using a microtome. The tissue sections were deparaffinized and stained with hematoxylin and eosin (H&E). Stained tissues were observed under a light microscope.

### 2.15. Statistical Analysis

Data are presented as the mean ± standard deviation (S.D.). Statistical analysis was performed using two-way analysis of variance and t-tests using Prism 8 (GraphPad Software, Inc., San Diego, CA, USA). * *p* < 0.05, ** *p* < 0.01 vs. control or between experimental groups.

## 3. Results and Discussion

### 3.1. Preparation and Physicochemical Characterization of iEDNs-DOX

The processes of anti-EGFR antibody coupling to the surface of EDNs, the encapsulation of DOX into anti-EGFR iEDNs, and the incorporation of QDs into the lipid bilayers of iEDNs are illustrated in Appendix A. Structurally stable EDNs were successfully prepared by sonication of the harvested erythrocyte ghost cells. The prepared EDNs had identical protein constituents as erythrocyte ghost membranes (Figure 1a), including CD47 molecules (Figure 1b). To prepare EGFR-targeting EDNs, thiol-maleimide reaction was used. The thiolated cetuximab antibodies were conjugated to maleimide moieties exposed on the surface of EDNs (Figure 1c,d), forming iEDNs. The detailed process for the thiol-maleimide reaction is explained in Section 2.4. Red cell ghosts, EDNs, iEDNs remained in the well of non-reducing gel (red box) while unconjugated antibodies were seen at the same position with free antibodies. According to the quantification of antibodies coupled to the EDNs, the antibody coupling efficiency was approximately 38%, and 114 μg of antibody was conjugated to 1 mg of EDNs. To optimize the DOX encapsulation into the iEDNs, the extravesicular buffer (25 mM HEPES) of iEDNs was replaced with 300 mM ammonium phosphate buffer to create a steep phosphate gradient. DOX was then added to the EDN solution followed by incubation for various periods. According to the encapsulated DOX measurement, approximately 38.5% of the drug was loaded into the EDNs after 48 h of incubation (Figure 1e). The encapsulation efficiency was higher than in previous reports [22]. The encapsulated DOX molecules were slowly released from the EDN vesicles over time (Figure 1f). According to the cumulative measurement of the DOX release from EDNs-DOX and iEDNs-DOX in a dialysis bag at 37 °C for 48 h, 62.5% of free DOX was released from the dialysis bag within 4 h, but only 7% and 3.5% of encapsulated DOX was released from EDNs and iEDNs, respectively, during the same time period. Even after 48 h, EDNs and iEDNs retained 39.6% and 44.9% DOX, respectively. These results suggest that EDN formulations can efficiently capture the DOX molecules inside and stably carry them.

The freshly prepared EDNs exhibited relatively homogeneous vesicular sizes and stable membranous integrity. According to the measurement of EDN diameter (Table 1), the size of EDNs (144.6 nm) was not significantly changed by DOX encapsulation or QD incorporation. However, the conjugation of anti-EGFR antibodies to the surface of EDNs increased the size to 171.2 nm. The surface charges of bare EDNs, EDNs-DOX, EDNs-QD, iEDNs, and iEDNs-DOX were −12.9, −14.5, −10.8, −12.5, and −13.9 mV, respectively. The processes of DOX encapsulation, QD incorporation, and antibody conjugation did not seriously affect the surface charge of EDNs, which was similar to that of erythrocyte plasma membranes.

TEM images of bare EDNs and other EDN derivatives (EDNs, EDNs-QDs, iEDNs, and iEDNs-DOX) also supported the EDN size measurements (Figure 2a). The integrity of EDNs was maintained after the loading of DOX and QDs and the coupling of the tumor-targeting antibodies. The sizes of EDNs appeared to be slightly increased by antibody conjugation but were not changed by DOX encapsulation. Negatively charged EDNs smaller than 180 nm in diameter would have advantages in circulation and penetration into tumor tissues owing to the enhanced permeability and retention (EPR) effect [23,24,25].

In general, it is known that the less stable the particles are, the more readily they are exposed to the immune system and then removed from the circulation due to aggregation with serum proteins [26,27]. To assess the in vitro stability of EDNs, they were stored in HEPES buffer in the absence or presence of serum proteins at 4 °C for 21 d, and their size changes were monitored with a DLS analyzer. The EDNs did not show any significant size changes, even in the presence of 50% FBS during this period (Figure 2b). This may imply that EDN formulations are structurally stable and biocompatible, making them suitable for in vivo administration.

### 3.2. Immune Surveillance Escape of EDNs

Since CD47 functions as a “don’t eat me” signal to phagocytic cells, it is reasonable to speculate that EDNs containing CD47 (Appendix A) would be taken up to a lesser extent by phagocytic immune cells, such as macrophages. The anti-phagocytic functions of EDNs were tested in murine macrophage cells using flow cytometry and confocal laser-scanning microscopy. First, the binding of freshly prepared EDNs to mouse macrophage RAW 264.7 cells was compared with that of EDNs treated with anti-CD47 antibody and EDNs treated with trypsin. These cell bindings were also directly compared to that of cationic lipid nanoparticles made of DMKE [28], which can non-specifically bind to the electronegative cell surface. After 1 h incubation at 37 °C, compared to the untreated control cells, the EDN-treated macrophages showed only 11.8% increase in mean fluorescence intensity (MFI), while the same cells treated with the CD47-blocked EDNs showed a 25.4% increase in MFI (Figure 3a). EDN damage caused by trypsin digestion showed a strong interaction with macrophages (62.9% MFI increase), which was comparable to the strong and non-specific interaction of DMKE cationic lipid nanoparticles (65.0% MFI increase). Freshly intact EDNs showed the lowest uptake into RAW 264.7 macrophages. However, CD47 blocking by anti-CD47 antibodies doubled the interaction with macrophages. This clearly shows that CD47 on the surface of EDNs interferes with the interactions with macrophages. The uncoating of EDNs by trypsinization removed most of the functional moieties of membrane proteins, including CD47, resulting in enhanced recognition by macrophages.

Microscopic analyses of the interactions between EDNs and macrophages also supported the reduced macrophage recognition of EDNs (Figure 3b). RAW 264.7 macrophages take up the trypsinized EDNs more readily than the intact ones. The EDNs were hardly seen in the cytoplasm of RAW 264.7 cells, but trypsinized-EDNs were heavily localized in the cytoplasm of macrophages. The results also imply that the CD47 molecules damaged by trypsinization do not function properly as immune surveillance signals. The scrutinized recognition of nanocarriers by immune cells has been the biggest hurdle to overcome for their clinical applications [29]. According to this result, incorporating CD47 molecules into nanocarriers as immune defense tools can aid in resolving the hurdle of immune recognition.

### 3.3. TNBC Cell-Targeted DOX Delivery by iEDNs

MDA-MB-231 and MDA-MB-453 breast cancer cells were used as in vitro EGFR-positive and EGFR–negative cancer models [30], respectively. Western blotting analyses using anti-EGFR antibody showed that MDA-MB-231 cells expressed a high amount of EGF receptors, but MDA-MB-453 cells did not (Appendix A). Before the cytotoxicity evaluation of iEDNs-DOX, the cytotoxicity of the tumor-targeting carrier itself was verified (Appendix A). The empty iEDNs did not show any toxicity to the two TNBC cell lines, up to 500 μg/mL of EDN proteins.

The iEDNs freshly prepared with Alexa Fluor 488-labeled anti-EGFR antibodies were used to treat MDA-MB-231 and MDA-MB-453 cells, and their interactions were analyzed by flow cytometry (Figure 4a). The treated MDA-MB-231 cells showed a stronger shift in MFI (53.36%) than MDA-MB-453 cells (3.89%). This implies that the iEDNs employing a tumor-targeting ligand, anti-EGFR antibody, were able to bind more specifically to the target MDA-MB-231 cells.

It is well established that DOX intercalation between dsDNA molecules results in cancer cell death [31]. The drug must be delivered to the cytoplasm of cancer cells. Therefore, it can be speculated that EGF receptor-mediated endocytosis may facilitate intracellular translocation of the drug from outside. To evaluate the tumor-specific delivery and distribution inside the cells, MDA-MB-231 and MDA-MB-453 cells were treated with three different DOX formulations: free DOX, EDNs-DOX, and iEDNs-DOX, and their cellular uptake was analyzed with a confocal microscope (Figure 4b). After 1 h of incubation, MDA-MB-231 cells treated with iEDNs-DOX showed higher DOX accumulation in the cytoplasm compared to the other two treatments. Meanwhile, untargeted MDA-MB-453 cells showed a low overall DOX signal. These results indicate that the anti-EGFR antibodies on the surface of EDNs can specifically recognize the target cells at an early stage and enhance the delivery of DOX to the cytoplasm.

However, the higher accumulation of iEDNs-DOX in the target cells did not enhance the in vitro cytotoxicity of the drug (Figure 4c). To evaluate the cytotoxicity of iEDNs-DOX, MDA-MB-231 and MDA-MB-453 cells were treated with various concentrations of free DOX, EDNs-DOX, and iEDNs-DOX for 24 h. According to the results of the CCK-8 assay, DOX encapsulation into EDNs noticeably reduced the cytotoxicity of the drug in MDA-MB-231 target cells. These phenomena have been logically explained by numerous previous reports [32,33]. In confined wells for cell culture, free drugs may have advantages in terms of accessibility and penetration into the cell via various endocytic pathways, such as pinocytosis, compared to vesicles encapsulating drugs. The IC_50_ values of free DOX, EDNs-DOX, and iEDNs-DOX were 0.02 μM, 0.18 μM, and 0.1 μM, respectively. The tumor-targeted iEDNs-DOX showed slightly stronger cytotoxicity than EDNs-DOX. Meanwhile, non-target MDA-MB-453 were less sensitive to free DOX (0.12 μM) and show similar levels of cytotoxic response to EDNs-DOX (0.48 μM) and iEDNs-DOX (0.71 μM).

### 3.4. Biodistribution of iEDNs-QD in Tumor-Xenografted Mice

QDs are semiconductor nanocrystals that have been utilized as molecular imaging agents [19,34]. To evaluate the theranostic applicability of iEDNs, QDs were incorporated into the bilayers of EDNs for bio-imaging (EDNs-QD) and tumor-specific antibodies were conjugated to EDNs-QDs (iEDNs-QDs). Then, MDA-MB-231-xenografted mice were intravenously injected with EDNs-QD or iEDNs-QD to evaluate the in vivo biodistribution of the particles. According to the real-time in vivo and ex vivo images (Figure 5a), the fluorescent signals of EDNs-QD and iEDNs-QD spread throughout the mouse body even just after injection. In particular, the iEDNs-QD signal of tumor tissues showed stronger fluorescence than the EDNs-QD at the location of tumor xenografts at 24 h and 48 h post-injection.

During imaging, ex vivo fluorescence images of major organs, such as the liver, lungs, heart, kidneys, and spleen, were taken as time elapsed (Figure 5b). Generally, the mice treated with the tumor-targeted iEDNs-QD showed stronger fluorescence from the tumor tissues than those treated with iEDNs-QD. The relative fluorescence intensity of the tumors increased with time and became the highest 48 h later (Figure 5c). Interestingly, the relative fluorescence intensity of blood was steadily maintained for 48 h after an approximately 60% decrease in the first 4 h post-injection (Figure 5d). QDs accumulation by tumors showed significant differences after 24 h injection (Figure 5e). This implies that the iEDNs were able to accumulate more effectively in the tumors than EDNs at the same time points. The increase in tumoral accumulation of iEDNs-QDs may be associated with effectively circulating particles. Presumably, a certain amount of the iEDNs properly exposes CD47 molecules and, therefore, circulates in the blood stably and effectively. Moreover, nanoparticles with tumor-sensing ligands and anti-EGFR antibodies may have a higher chance of recognizing tumor cells during repeated blood circulation through tumor tissues. Compared to the hepatic accumulations of other synthetic nanoparticles [35], the iEDNs and EDN showed relatively low levels of hepatic uptake. This also implies that the hepatic reticuloendothelial system does not effectively remove iEDNs from the blood because of the immune-avoiding capability of the particles.

Numerous QDs have been suggested as a bioimaging reagent for cancers [36], but there are very limited reports regarding delivery systems for the QDs [37]. Previously, we have suggested two different types of QD carriers. Lipid micelles encapsulating QDs and paclitaxel were coupled to anti-EGFR antibodies and then tested in a mouse cancer [19]. In addition, anti-cancer therapeutic siRNA and QDs were co-captured in lipid nanocarriers and tested in mice carrying cancers [34]. In this study, the biodistribution patterns of iEDNs carrying QDs showed an effective delivery to target tumors and less hepatic uptake, implying that the iEDNs are also suitable delivery systems for tumor-directed theranosis.

### 3.5. Anti-Cancer Therapeutic Effects of iEDNs-DOX

To evaluate the in vivo anti-cancer therapeutic efficacy of iEDNs-DOX, free DOX, EDNs-DOX, and iEDNs-DOX were intravenously injected into MDA-MB-231-xenografted mice, and their tumor growth was compared with that of saline-treated mice (Figure 6). According to the tumor growth measurement for 32 d, the mice treated with DOX formulated in erythrocyte membranes exhibited slower tumor growth than those treated with free DOX (Figure 6a). Between the two EDN formulations of DOX, tumor-targeted iEDNs-DOX showed stronger effectiveness in inhibiting tumor growth than EDNs-DOX, which was more statistically meaningful as time elapsed. The average tumor sizes of iEDNs-DOX-treated tumors at the start point (day 1) and end point (day 32) were 95 and 200 mm^3^, respectively. After the last measurement, the tumors were resected from all mice and weighed. The gross appearance of the tumors showed that iEDNs-DOX was the most effective in inhibiting tumor growth, followed by EDN-DOX (Figure 6b). The direct measurement of tumor weight statistically verified the therapeutic effectiveness of iEDNs-DOX (Figure 6c). The average tumor weights of the mice treated with saline, free DOX, EDNs-DOX, and iEDNs-DOX were 0.42, 0.22, 0.15, and 0.06 g, respectively. These results also suggest that anti-EGFR iEDNs are efficient vehicles for cancer-targeted drug delivery.

For the in situ assessment of apoptotic tumor cells, the tumors dissected from the mice injected with saline, free DOX, EDNs-DOX, or iEDNs-DOX were subjected to the TUNEL assay. TUNEL-positive cells in the tumors of mice injected with saline, free DOX, EDNs-DOX, and iEDNs-DOX were 8.3 ± 2.6, 14.5 ± 5.5, 77.3 ± 28.7, and 148.0 ± 14.1 cells per 600 counts (Figure 6d). The tumors treated with saline and free DOX showed fewer TUNEL-positive cells. Among the treatment groups, iEDNs-DOX had the highest number of apoptotic cells in the tumor tissues in situ. It has been suggested that general H&E staining can reveal the infiltration of immune cells and fibrosis resulting from the immune responses after anti-cancer drug administration [38,39]. According to our H&E staining of the tumor sections, a large number of immune cells were also localized in all the tumor tissues, resulting in severe fibrosis (Figure 6d).

To evaluate the aberrant in vivo toxicity of DOX formulated EDNs, the body weights of mice treated with free DOX, EDNs-DOX, or iEDNs-DOX were measured throughout the experiment. In fact, there was no serious bodyweight loss resulting from any abnormal toxicity of the DOX formulations (Figure 6e). In addition, physical abnormalities in the treated mice were also not monitored.

To elucidate any histopathological damage resulting from the repeated treatments with various DOX formulations, major organs, such as the liver, heart, lungs, spleen, and kidneys, were dissected from the treated mice and stained with H&E (Figure 7). No obvious pathological abnormalities were observed in any of the organs examined compared with the saline-treated control group, indicating the clinical safety of the EDN formulations. The analyses of in vivo toxicity and histopathology imply that iEDNs are biocompatible and safe vehicles for the co-delivery of anti-cancer drugs and bioimaging reagents.

There are very few reports regarding theranostic approaches using QDs and anticancer drugs. The novel nanostructured lipid carriers co-formulating QDs and paclitaxel exhibited enhanced anti-cancer therapeutic efficacy and safety, compared to free paclitaxel [37]. The anti-EGFR iEDNs in this study are further renovative carriers in terms of biocompatibility and active tumor targeting. The iEDN system may have stronger merit in clinical applications.

## 4. Conclusions

In this study, EGFR-targeted EDNs were developed for the targeted treatment of TNBC. The integrity of prepared EDNs is little affected by DOX encapsulation and QD incorporation. The EDNs had ample cargo efficiency as a delivery system. Therefore, a variety of anti-cancer drugs and bioimaging molecules can be loaded into the EDNs for cancer-targeted theranosis. The coupling of anti-EGFR antibody molecules on the surface of EDNs (iEDNs) enhanced the delivery efficacy of anticancer drugs and QDs loaded in the EDNs. The iEDNs-DOX showed a far better anticancer effect in vivo than the untargeted version. A variety of different tumor-targeting antibodies can be adapted to the iEDN system. In addition, various anticancer therapeutics can be also loaded in the iEDNs. Especially, QDs were successfully incorporated into the erythrocyte membrane bilayers for tumor bioimaging. The anti-EGFR iEDNs-QD showed the most effective tumor imaging for 48 h after injection. In summary, the anti-EGFR iEDNs could deliver DOX and QDs simultaneously and efficiently to target tumor tissues. These results suggest that this novel anti-cancer theranostic vehicle derived from erythrocyte plasma membranes with tumor-targeting antibodies can be utilized for targeted theranosis of cancers.

## Figures and Tables

**Figure 1 pharmaceutics-15-00350-f001:**
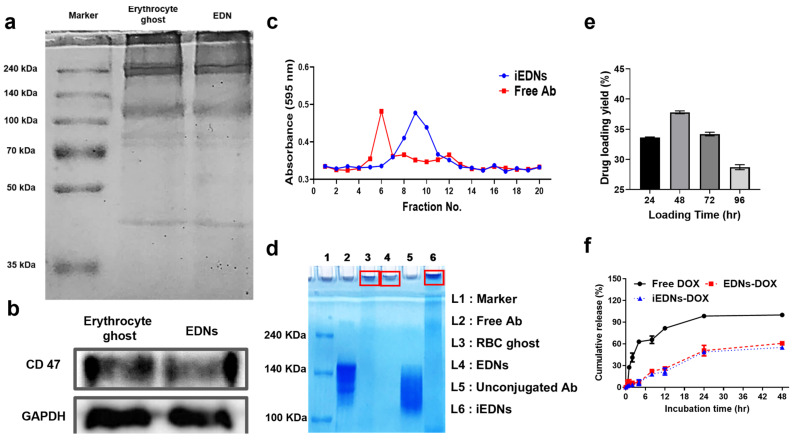
Preparation and characterization of immuno-erythrocyte-derived nanoparticles (EDNs) (iEDNs)-doxorubicin (DOX). (**a**) Membrane proteins in erythrocyte ghosts and EDNs were separated by 8% sodium dodecyl sulfate-polyacrylamide gel electrophoresis (SDS-PAGE) and stained with Coomassie Brilliant Blue. (**b**) The expression of cluster of differentiation CD47 of Erythrocyte ghost and EDNs was verified by Western blotting. The expression of GAPDH is used for control. (**c**) After cetuximab conjugation to the EDN surface, iEDNs were separated by gel filtration and (**d**) the separated iEDNs were subjected to 6% non-reducing PAGE. (**e**) For DOX encapsulation, the drug was added to the iEDNs that had 300 mM ammonium phosphate inside and 25 mM 4-(2-hydroxyethyl)-1-piperazineëthanesulfonic acid (HEPES) outside, and incubated. Time-dependent DOX encapsulation into the iEDNs was analyzed by measuring the absorbance at 480 nm. (**f**) While dialyzing EDNs-DOX, the released DOX was quantified at various time points. Each error bar represents the mean ± standard deviation (S.D.) for three separate experiments.

**Figure 2 pharmaceutics-15-00350-f002:**
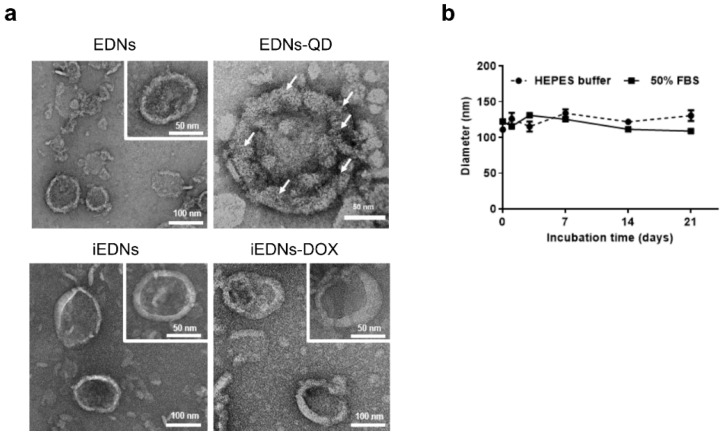
Changes in the morphologies and sizes of EDNs. (**a**) Morphologies of freshly prepared EDNs, EDNs-quantum dots (QDs), EDNs-DOX, and iEDNs-DOX were observed by transmission electron microscopy (TEM). The white arrows indicate QDs. (**b**) Sizes of EDNs in serum-free HEPES or 50% fetal bovine serum (FBS) were measured for 21 d of storage at 4 °C.

**Figure 3 pharmaceutics-15-00350-f003:**
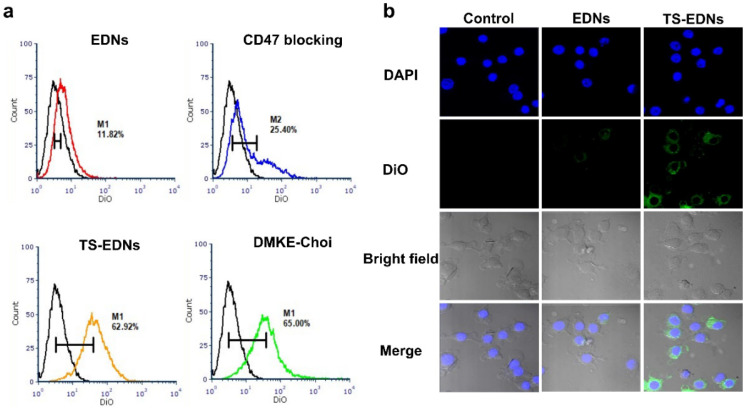
Interactions of EDNs with macrophages. Freshly prepared EDNs, CD47-blocked EDNs, trypsinized (TS) EDNs, and O,O′-dimyristyl-*N*-lysyl glutamate (DMKE)-based cationic lipid nanoparticles were labeled with DiO. RAW 264.7 mouse macrophage cells were then treated with fluorescence-labeled nanoparticles for 1 h at 37 °C and subjected to flow cytometry (**a**) and confocal microscopy (**b**). Magnification: ×400.

**Figure 4 pharmaceutics-15-00350-f004:**
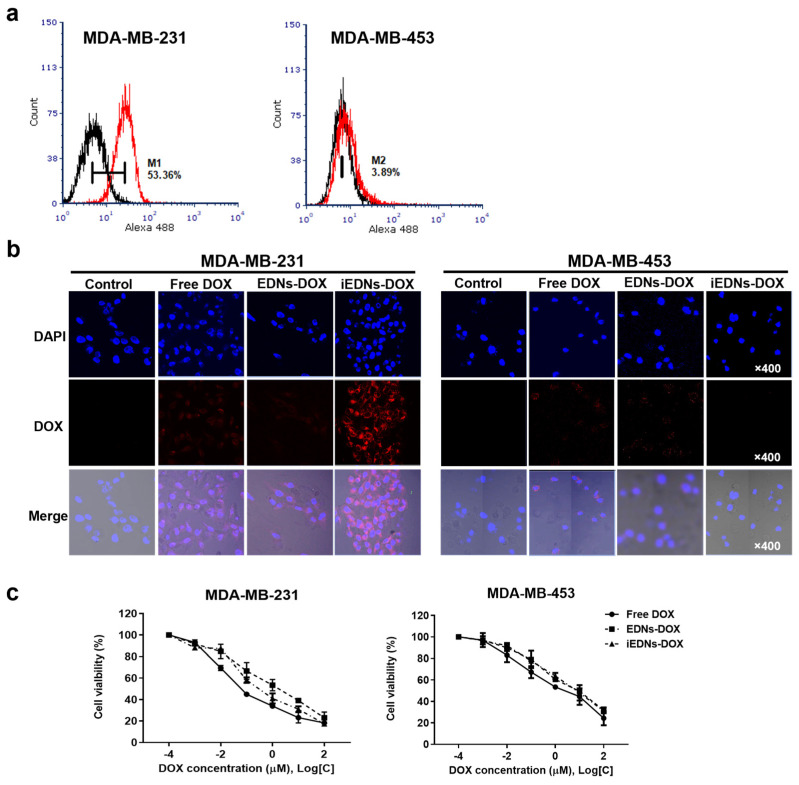
Tumor-targeted intracellular drug delivery and anti-cancer effects of iEDNs-DOX. (**a**) EGFR+ MDA-MB-231 and EGFR− MDA-MB-453 cells were treated with Alexa Fluor 488-labeled iEDNs for 1 h at 4 °C and analyzed by flow cytometry. (**b**) Cancer cells were treated with free DOX, EDNs-DOX, or iEDNs-DOX at 37 °C for 1 h and analyzed by confocal microscopy with DOX fluorescence in the RFP channel. (**c**) Cancer cells were treated with free DOX, EDNs-DOX, or iEDNs-DOX for 24 h and the viabilities of the treated cells were measured using the cell counting kit-8 (CCK-8) assay. Each error bar represents the mean ± S.D. for three separate experiments (n = 3).

**Figure 5 pharmaceutics-15-00350-f005:**
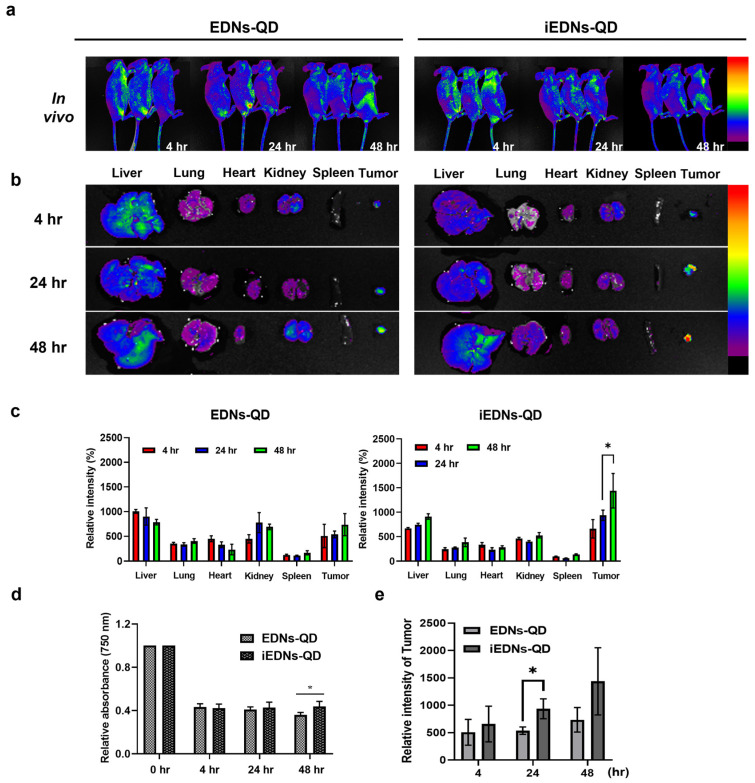
In vivo and ex vivo analyses of the biodistribution of iEDNs-QDs. (**a**) Tumor-xenografted mice were intravenously (i.v.) injected with EDNs-QDs or iEDNs-QDs and their whole bodies were imaged at 4, 24, and 48 h post injection. (**b**) During the whole-body imaging, major organs and tumors were dissected and their ex vivo fluorescence signals were imaged. (**c**) Relative QD fluorescence of the organs and tumors were measured. (**d**) Relative QD fluorescence of the blood taken from the mice treated with EDNs-QDs or iEDNs-QDs was measured at 0, 4, 24, and 48 h post injection and relative intensity of tumor tissues in (**e**). Each error bar represents the mean ± S.D. for three mice. * *p* < 0.05.

**Figure 6 pharmaceutics-15-00350-f006:**
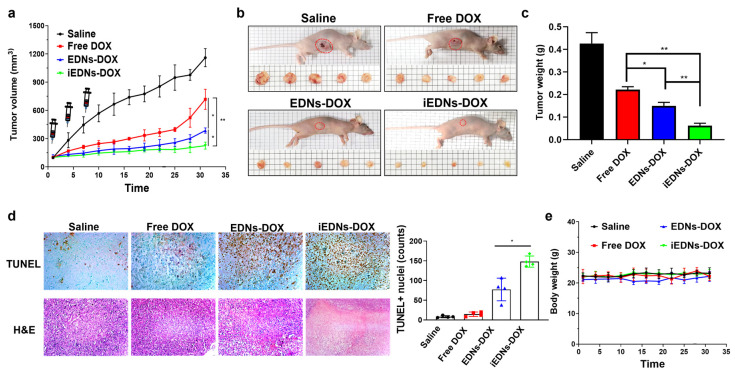
In vivo tumor growth inhibition by iEDNs-DOX. MDA-MB-231-xenografted mice (n = 5) were i.v. injected with saline, free DOX, EDNs-DOX, or iEDNs-DOX (4 mg/kg of DOX) thrice at day 1, 4, and 7. (**a**) Tumor sizes and body weights were measured thrice every week for 32 d. (**b**) At the end point, the gross appearance of the resected tumors was observed and (**c**) the tumors were weighed. (**d**) Apoptotic cells in the sections of tumors were counted by the terminal deoxynucleotidyl transferase dUTP nick and labeling (TUNEL) assay and visualized after hematoxylin and eosin (H&E) staining. (**e**) The bodyweights of the treated mice were measured thrice every week. Each error bar represents the mean ± S.D (n = 5). * *p* < 0.05 and ** *p* < 0.01.

**Figure 7 pharmaceutics-15-00350-f007:**
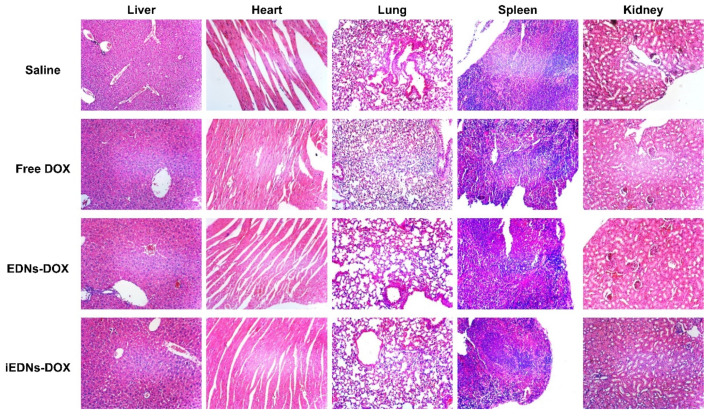
Histopathological examination of major internal organs of the mice injected with iEDNs-DOX. Major organs, including the liver, heart, lungs, spleen, and kidneys, were dissected from the treated mice on day 32 after the administration of various DOX formulations. The paraffin-embedded tissue sections of the organs were stained with H&E and observed under a light microscope. Magnification: ×100.

**Table 1 pharmaceutics-15-00350-t001:** Physicochemical characterization of erythrocyte-derived nanoparticles (EDNs).

	Size ^a^ (nm)	Polydispersity Index ^a^ (PDI)	Zeta Potential ^a^ (mV)	DOX Encapsulation Efficiency (%)
EDNs	144.6 ± 8.5	0.154 ± 0.025	−12.9 ± 1.2	-
Immuno-EDNs	171.2 ± 11.6	0.170 ± 0.025	−12.5 ± 0.8	-
EDNs-QD	147.6 ± 10.5	0.342 ± 0.015	−11.4 ± 1.1	-
DOX-EDNs	146.5 ± 11.4	0.298 ± 0.028	−14.5 ± 1.3	37.8 ± 1.5%
Immuno-EDNs-DOX	168.3 ± 15.3	0.247 ± 0.009	−13.9 ± 0.7	35.2 ± 2.6%

^a^ Diameter and Zeta potential were measured three times with a particle analyzer.

## Data Availability

Data are available upon request.

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
