# Peer review of "Tumor-Targeted Erythrocyte Membrane Nanoparticles for Theranostics of Triple-Negative Breast Cancer"

_pharmaceutics, 2023, doi:10.3390/pharmaceutics15020350_

Round 1

Reviewer 1 Report (New Reviewer)

The paper is aimed at developing triple-negative breast cancer targeted co-delivery systems for therapeutics and diagnostics. This work contains results of multiple physical methods collected together in order to give comprehensive view for the audience of Pharmaceutics. The paper is clear, conclusions are supported by correctly carried out experimental results, so recommendation is accept in the present form. 

Author Response

Reviewer #1

The paper is aimed at developing triple-negative breast cancer targeted co-delivery systems for therapeutics and diagnostics. This work contains results of multiple physical methods collected together in order to give comprehensive view for the audience of Pharmaceutics. The paper is clear, conclusions are supported by correctly carried out experimental results, so recommendation is accept in the present form. 

: We appreciate the comments.

Reviewer 2 Report (New Reviewer)

Theranostic approaches using QDs and anticancer drugs are rarely reported. Using EGFR as therapeutic target is a good idea, but its wide distribution in the body should be considered, whether there will be additional side effects, such as diarrhea caused by targeting EGFR. The new form of effective drugs is expected to enhance the efficacy and reduce the toxicity. For the new form, it may affect its distribution in different organs. Also whether other ingredients have additional side effects is more concerned. So I would like to see more detailed parameters about the toxicity, such as ID50 etc.

Author Response

Minor revision

pharmaceutics-2082930

Reviewer #2

Theranostic approaches using QDs and anticancer drugs are rarely reported. Using EGFR as therapeutic target is a good idea, but its wide distribution in the body should be considered, whether there will be additional side effects, such as diarrhea caused by targeting EGFR. The new form of effective drugs is expected to enhance the efficacy and reduce the toxicity. For the new form, it may affect its distribution in different organs. Also whether other ingredients have additional side effects is more concerned. So I would like to see more detailed parameters about the toxicity, such as ID50 etc.

: We appreciate the comments. According to the histological analysis (H&E) of major organs, any aberrant changes resulting from the non-specific side effect of the drug formulations were not monitored. Even, the carriers (EDNs and iEDNs) did not exhibit any cellular toxicity to the target cells at various concentrations (Fig S4).

Reviewer 3 Report (New Reviewer)

The manuscript by Choi et al. demonstrates the rational design and successful synthesis of erythrocyte-derived nanoparticles (EDNs) encapsulating doxorubicin (DOX) and quantum dots (QDs). The resulting anti-EGFR iEDNs exhibited strong biocompatibility, prolonged blood circulation, and efficient targeting of TNBC in mice.

Overall, this study is well designed and performed for the materials and functional biological studies. The obtained results could support the conclusion. Therefore, I believe that this research is acceptable for publication in Pharmaceutics.

Besides, minor revisions should be made to enhance the quality of this paper.

1. Some fonts in images of the manuscript look small. Please enlarge them.

2. The image arrangement is a little messy. Please revise them to make the images look regular.

3. Polydispersity index of some particles encapsulating QDs looks abnormally high, leading to the instability concern for those particles. Please clarify it.

Author Response

pharmaceutics-2082930

Reviewer #3

The manuscript by Choi et al. demonstrates the rational design and successful synthesis of erythrocyte-derived nanoparticles (EDNs) encapsulating doxorubicin (DOX) and quantum dots (QDs). The resulting anti-EGFR iEDNs exhibited strong biocompatibility, prolonged blood circulation, and efficient targeting of TNBC in mice.Overall, this study is well designed and performed for the materials and functional biological studies. The obtained results could support the conclusion. Therefore, I believe that this research is acceptable for publication in Pharmaceutics.

Besides, minor revisions should be made to enhance the quality of this paper.

  1. Some fonts in images of the manuscript look small. Please enlarge them.

: As suggested, the font size in Figures 1 and 6 was amended.

  1. The image arrangement is a little messy. Please revise them to make the images look regular.

: As suggested, some image arrangements were amended.  

  1. Polydispersity index of some particles encapsulating QDs looks abnormally high, leading to the instability concern for those particles. Please clarify it.

: We appreciate the comments. The PDI of EDNs-QD was somehow higher compared to those of the other particles. However, the number (0.342) may be still in the acceptable range of PDI (<0.4) for nanomedicine applications (Drug Delivery, 2021, 28:973).

This manuscript is a resubmission of an earlier submission. The following is a list of the peer review reports and author responses from that submission.

Round 1

Author Response

Responses to reviewers’ comments

Manuscript ID: pharmaceutics-2010806                           # Reviewer1

Dear Reviewer,

Enclosed please find files of the amended manuscript and a letter of response to the referees. Comments are to be addressed below. I appreciated. 

Response to comments:

-Figure 1d: please add molecular weight designations for the protein standard marker lane; it would also be helpful for reviewers to see the entire gel. (Where are EDNs and iEDNs expected to be in the gel?)

à Molecular weights of markers were added to Figure 1d. EDNs and iEDNs were in the wells of non-reducing PAGE, which was mentioned in the revised manuscript (Line 276-278).

-Lines 109-110 state “Uncaptured QD aggregates were removed by filtering through a 100 nm polycarbonate membrane filter”. What size are the QDs? Considering the size of the EDNs and iEDNs (>140 nm), large aggregates would likely be present with the EDNs after filtering. Thus this technique may not be sufficient to remove aggregates from the desired EDNs.

à The size of a QD is about 5 nm. However, the hydrophobicity of the QDs makes the uncaptured ones aggregate with each other and then precipitate. The uncaptured QDs must be removed from captured ones (smaller than 100 nm) before mixing with red cell ghosts.

-Please indicate in the caption of Figure 2 what the arrows represent in Figure 2a for the “EDNsQD” image.

à Explanation of the arrows is included in Figure 2a.

 -Figure 5c: it is recommended to use the same scale on the vertical axis for both EDNs and iEDNs graphs. Also, for the scale on the vertical axis, please clarify what the % relative intensity is compared to, and why the numbers shown are as high as 1500%?

à The mistakes in the Y legends were corrected in Figure 5c.

-Please clarify whether or not a pathologist reviewed the tissue sections to analyze potential toxic effects of the EDNs or iEDNs relative to DOX?

à Our department (Biomedical Lab. Science) is specialized in pathological examination and analyses.  

Additional comments: -It is not clear if “theranostic” EDNs were prepared or used in these studies, because it seems that EDNs containing the QDs for imaging were separate from EDNs containing the drug for therapy. Please clarify or explain this point in the discussion.

à The in vivo bioimaging system used in this study was not able to clearly distinguish the fluorescence from doxorubicin and QD because their emission wavelengths are largely superimposed. Therefore, we had to proceed with the biodistribution assay and tumor growth inhibition assay using EDNs-DOX and EDNs-QDs, respectively.  

-Please ensure that superscript and subscript formatting is correct throughout the manuscript (especially Section 2). -Section 2.4 states 6% SDS-PAGE was used to characterize iEDNs, and section 2.7 states 10% SDS-PAGE was used, but Figure 1 caption (lines 457 and 460) states 8% SDS-PAGE was used; please clarify these discrepancies.

à The mistakes in the superscript and subscript were corrected. The SDS-PAGE analyses of EDNs, Ab-purification, and western blotting were done on 8%, 6% and, 10% gel, respectively. Those was indicated the legend of Figure 1.    

-Line 310: recommend changing “in vivo” to “in vitro” since the stability was tested in solution.

à As suggested, the expression was removed (Line 313).

-Line 347: recommend changing the heading for Section 3.1.3, because discussion was restricted to results using tumor cells in vitro (not tumors).

à Amended as suggested (Line 350).

-Line 398: recommend deleting “During real-time imaging” (real-time imaging normally refers to a live recording or video, not to a static image at an isolated time point).

à Amended as suggested (Line 401)

-Line 519: recommend deleting “in vitro and” (because the in vitro results did not show increased cytotoxicity of EDNs relative to free drug, as the authors state in Section 3.1.3).

à Amended as suggested (Line 522)

-Please review the Conclusion (Section 4) paragraph to revise grammar and wording.

à The conclusion was replaced with a new version.  

-While the results from the experiments are presented and discussed, very little “discussion” is presented for the findings of the study relative to other existing literature. Significantly more discussion that would be of interest to the scientific community should be added, considering the findings from these studies.

à As requested, more discussion was included (Line 418-426, Line 465-470).

Reviewer 2 Report

Tumor-targeted erythrocyte membrane nanoparticles for theranostics of triple-negative breast cancer

Summary: The authors describe erythrocyte derived nanoparticles loaded with doxorubicin, quantum dots and functionalized with an EGFR targeting antibody for delivering drug and imaging agents to triple negative breast cancer. The manuscript is descriptive enough, however clarity is missing in few aspects, and will benefit from statistics at multiple places. The manuscript under consideration requires following improvements:

Major comments:

1)    Clearly define EDNs in the manuscript. Are EDNs referring to erythrocyte ghosts only, or erythrocyte ghosts loaded with DSPE-PEG-QD micelles? At different places in the manuscript, the definition for EDNs is used interchangeably, maintain consistency. Similarly, clearly describe all components of various nanoparticles used in the manuscript: for example, would iEDNs-DOX have QDs as well? Maintain and define consistent nomenclature throughout the manuscript.

2)    QDs not encapsulated in micelles are <10 nm in size. Was there any confirmation done to ensure all QDs are associated with PEG-DSPE micelles?

3)    Line 276: data missing for antibody conjugation to EDNs. Explain and include data (can be supplemental) for how this was performed.

4)    Lines 278-280: mismatch in text compared to description in methods section

5)     Figure 1a and d: these gels cannot be interpreted, lots of sample is stuck in the wells as shown by blue staining in the wells. Move them to supplemental since they cannot be interpreted

6)    Figure 1c: Explain how free Ab has absorbance at 595 nm? 

7)    Figure 1e: explain the observed trend, why does loading yield go down at longer times?

8)    Figure 1f: include statistics, fit to appropriate kinetic models and include fit lines, 95% CI, and half-lives of release, perform tests for significance across groups.

9)    Figure 2a: Were the experiments performed in replicates? For flow data, include pairwise comparisons for MFI and identify which differences are statistically significant.

10) Figure 2b: Include 2 more panels: CD47 clocked EDNs and DMKE-Chol. Draw ROIs around cells, measure integrated fluorescence intensities, perform pairwise comparisons by one-way ANOVA

11) Figure 4b: Is the signal in red channel from native fluorescence of DOX? Please clarify in text

12) Figure 4c: the concentration ranges for dose-response curves need to be expanded to achieve full sigmoidal curve. Fit dose-response curves to appropriate models, include 95% CI of the fit lines, measure IC50 in triplicates and perform pair-wise statistics on IC50 to identify which groups are significantly different from each other.

13) Figure 5a: Normal mice can sometimes have high background due to florescent molecules in their diet etc. Were any normal (untreated) mice included in the bioimaging study. If so, include the data.

14) Figure5 c: y-axis is labeled as relative intensity (%), however, the axis number do not range from 0-100. 

15) Figure 5c: Improve statistical analysis. Statistical comparisons between EDNs and iEDNs are also important to measure the efficiency of targeting

16) Figure 6a: Was a vehicle only control arm included? Saline is not the best negative control.

Minor comments:

1)    More details on the QDs used should be included in materials section (include its optiocal properties).

2)    Is the procedure for EDN, iEDN preparation and DOX encapsulation published before? Provide multiple literature references for the same

3)    In lines 107-114, the mass ratio of DSPE-PEF-QD : erythrocyte ghosts is missing

4)    Figure S2: expand the figure to include earlier synthesis steps as well (QD micelle generation etc.)

5)    Include data and equations used to measure DOX encapsulation

6)    Figure 4a: labels missing

7)    Grammar check is required for conclusions section

Author Response

Responses to reviewer’s comments

Manuscript ID: pharmaceutics-2010806

#Reviewer 2

Tumor-targeted erythrocyte membrane nanoparticles for theranostics of triple-negative breast cancer

Summary: The authors describe erythrocyte derived nanoparticles loaded with doxorubicin, quantum dots and functionalized with an EGFR targeting antibody for delivering drug and imaging agents to triple negative breast cancer. The manuscript is descriptive enough, however clarity is missing in few aspects, and will benefit from statistics at multiple places. The manuscript under consideration requires following improvements:

Major comments:

1)    Clearly define EDNs in the manuscript. Are EDNs referring to erythrocyte ghosts only, or erythrocyte ghosts loaded with DSPE-PEG-QD micelles? At different places in the manuscript, the definition for EDNs is used interchangeably, maintain consistency. Similarly, clearly describe all components of various nanoparticles used in the manuscript: for example, would iEDNs-DOX have QDs as well? Maintain and define consistent nomenclature throughout the manuscript.

à EDN is only from the erythrocyte ghost but EDNs-QD is ghosts loaded with DSPE-PEG-QD micelle. In section 2.4, I mentioned the whole process of preparation for each particle.

2)    QDs not encapsulated in micelles are <10 nm in size. Was there any confirmation done to ensure all QDs are associated with PEG-DSPE micelles?

à The size of QDs is <5 nm and has a hydrophobicity. Because the inside of the micelle is also hydrophobic, QDs could be encapsulated into the inner space of the micelle. The size of QD-micelle is 50-70 nm. And our lab has used QD-micelle in other research on the same condition and same preparation. And I added the additional reference about QD micelle to Ref 31.  

3)    Line 276: data missing for antibody conjugation to EDNs. Explain and include data (can be supplemental) for how this was performed.

à I mentioned the whole process of antibody conjugation in Supplement Fig2. And in figure 1d, the conjugation of Abs was confirmed.

4)    Lines 278-280: mismatch in text compared to description in methods section

à The method for Ab conjugation, I added the 300 μg of cetuximab to 1 mg EDNs and 114μg of cetuximab bound to the particles. So the conjugation efficiency is about 38%.

5)     Figure 1a and d: these gels cannot be interpreted, lots of sample is stuck in the wells as shown by blue staining in the wells. Move them to supplemental since they cannot be interpreted

à I corrected the manuscript as you recommend.

6)    Figure 1c: Explain how free Ab has absorbance at 595 nm? 

à It had some labeling errors. The red line is for iEDNs and blue is results of Free Abs. And I used the Bradford reagents so the wavelength of absorbance is 595 nm.

7)    Figure 1e: explain the observed trend, why does loading yield go down at longer times?

à After 48hr of encapsulation, the protein of EDNs aggregation. And increasing the incubation time, the phosphate gradient was decreasing. These two reasons affected to encapsulation efficiency of DOX.   

8)    Figure 1f: include statistics, fit to appropriate kinetic models and include fit lines, 95% CI, and half-lives of release, perform tests for significance across groups.

à I added the statistics.

9)    Figure 2a: Were the experiments performed in replicates? For flow data, include pairwise comparisons for MFI and identify which differences are statistically significant.

à For figure 3a, I proceeded with single samples. 

10) Figure 2b: Include 2 more panels: CD47 clocked EDNs and DMKE-Chol. Draw ROIs around cells, measure integrated fluorescence intensities, perform pairwise comparisons by one-way ANOVA

à Unfortunately, I have no access to that machine and software anymore. I couldn’t additional data set.

11) Figure 4b: Is the signal in red channel from native fluorescence of DOX? Please clarify in text

à Yes. This red signal was from original DOX. I added the mention in figure caption.

12) Figure 4c: the concentration ranges for dose-response curves need to be expanded to achieve full sigmoidal curve. Fit dose-response curves to appropriate models, include 95% CI of the fit lines, measure IC50 in triplicates and perform pair-wise statistics on IC50 to identify which groups are significantly different from each other.

à I already checked the statistics and IC50. And about this, I mentioned revised manuscript in line 383.

13) Figure 5a: Normal mice can sometimes have high background due to florescent molecules in their diet etc. Were any normal (untreated) mice included in the bioimaging study. If so, include the data.

à Because this biodistribution is for the comparing targetability between EDNs and iEDNs, i didn’t set up the PBS group. EDNs-QD was used for baseline of this experiment.

14) Figure5 c: y-axis is labeled as relative intensity (%), however, the axis number do not range from 0-100. 

à I corrected.

15) Figure 5c: Improve statistical analysis. Statistical comparisons between EDNs and iEDNs are also important to measure the efficiency of targeting

à Thank you for your point. I added the graph to Fig 5d. At 24hr after injection, EDNs and iEDNs showed significant differences.

16) Figure 6a: Was a vehicle only control arm included? Saline is not the best negative control.

à Unfortunately, there was not a vehicle only. The free DOX group used for control.

Minor comments:

1)    More details on the QDs used should be included in materials section (include its optiocal properties).

à I added the additional mention about QDs in line 418-426.

2)    Is the procedure for EDN, iEDN preparation and DOX encapsulation published before? Provide multiple literature references for the same

à Some researched about erythrocyte-camouflaged particles but bare membrane nanoparticle is few. The ref 14, 15, and 19.

3)    In lines 107-114, the mass ratio of DSPE-PEF-QD : erythrocyte ghosts is missing

à It was mentioned in lines 104-105

4)    Figure S2: expand the figure to include earlier synthesis steps as well (QD micelle generation etc.)

à Detailed process about the QD micelle was described in ref 31.

5)    Include data and equations used to measure DOX encapsulation

à It was mentioned in section 2.4.

6)    Figure 4a: labels missing

à I corrected

7)    Grammar check is required for conclusions section

à I did. Thank you

Reviewer 3 Report

The manuscript (pharmaceutics-2010806) provides a sound description and interesting investigation, although the following suggestions will be further helpful to improve its lacking. 

1. The present form of abstract provides generalize overview of the work, in my opinion finding should also be incorporated in revised manuscript in a form of absolute results.

2. Authors have to incorporate the formulation related literature carried by other researcher and highlight the current research gap. It is suggested to link the current investigation how helpful to fulfil the formulation related research gap including merits of your work.

3. The encapsulation efficiency of DOX-EDNs and Immuno-EDNs-DOX are 37.8 ± 1.5 % and 35.2 ± 2.6 % respectively, which is low. In my opinion, effort has to be discussed in revised manuscript to increase its encapsulation efficiency. It is also suggested to incorporate the reason for its low encapsulation efficiency compared to other nano carrier system exploited in drug delivery in cancer.

4. It is suggested to include the dose of drug (mg/kg) and amount of formulation (in a form of volume) administered to the animals in in vivo study (Biodistribution analysis, tumor growth inhibition study). This information should be in detail in corresponding section 2.11 and 2.12 respectively.

5. The discussion should be improved particularly of section 3.1.5 “Anti-cancer therapeutic effects of iEDNs-DOX”. Author has to highlight/label the different structures in the tissues of liver, heart, lungs, spleen, and kidney to show no changes observed on those labeled structured of the tissues and observed similar to normal tissues, or if any changes observed compared to normal tissues then it should be incorporated to improve the discussion of histopathology investigations.  Similarly, discussion should be improved for in vivo antitumor investigation as well. 

Author Response

Responses to reviewers’ comments

Manuscript ID: pharmaceutics-2010806                                       # Reviewer 3

Dear Reviewer,

Enclosed please find files of the amended manuscript and a letter of response to the referees. Comments are to be addressed below. I appreciated. 

Response to comments:

- In title, there is some unnecessary firstname lastname words are there, which should be correct and properly checked.

à Corrected as suggested.

-In introduction, authors should a bit more focus on EGFR receptors highly expressed in TNBC with more literature support.

à More literature was included. I added additional references.

- Authors mentioned that QDs are hydrophobic, need to focus on their biocompatibility and some literature say QDs are hydrophilic. It is recommended to double check.

à We have previously reported several studies regarding the novel delivery systems for hydrophobic QDs (ref. 30,31). We are sure of the physicochemical properties of QDs.

-Results and Discussion sections need rearrangement. All the figures are at the bottom of the results and discussion, so, try to fit the figures after each section which will make easy to read results and to look at figures.

à As far as we know, the present format is temporary, and all the figures and a table will be assigned to appropriate positions.

-Data look very promising but authors did not discuss well with support from previous literature.

à As requested, more discussion was included (Line 418-426, Line 465-470).

Reviewer 4 Report

In the manuscript by Choi et al, Tumor-targeted erythrocyte membrane nanoparticles for theranostics of triple-negative breast cancer in which authors have developed novel tumor targeted nanoparticles loading with anti-tumor drug and imaging agent for theranostic applications. This manuscript tried to solve the current unmet need for TNBC, and the manuscript is well designed and executed with experimental design. Some of the comments and suggestions to be addressed before publications are given below:

In title, there is some unnecessary firstname lastname words are there, which should be correct and properly checked.

-In introduction, authors should a bit more focus on EGFR receptors highly expressed in TNBC with more literature support.

- Authors mentioned that QDs are hydrophobic, need to focus on their biocompatibility and some literature say QDs are hydrophilic. It is recommended to double check.

-Results and Discussion sections need rearrangement. All the figures are at the bottom of the results and discussion, so, try to fit the figures after each section which will make easy to read results and to look at figures.

-Data look very promising but authors did not discuss well with support from previous literature.

Author Response

Responses to reviewers’ comments
Manuscript ID: pharmaceutics-2010806
# Reviewer 3
In title, there is some unnecessary firstname lastname words are there, which should be correct and 
properly checked.
→ Corrected as suggested. 
-In introduction, authors should a bit more focus on EGFR receptors highly expressed in TNBC with 
more literature support.
→ More literature was included. I added additional references.
- Authors mentioned that QDs are hydrophobic, need to focus on their biocompatibility and some 
literature say QDs are hydrophilic. It is recommended to double check.
→ We have previously reported several studies regarding the novel delivery systems for hydrophobic 
QDs (ref. 30,31). We are sure of the physicochemical properties of QDs. 
-Results and Discussion sections need rearrangement. All the figures are at the bottom of the results and 
discussion, so, try to fit the figures after each section which will make easy to read results and to look 
at figures.
→ As far as we know, the present format is temporary, and all the figures and a table will be assigned 
to appropriate positions.
-Data look very promising but authors did not discuss well with support from previous literature.
→ As requested, more discussion was included (Line 418-426, Line 465-470).

Round 2

Reviewer 2 Report

Based on the responses provided by the authors to my previous comments, I do not recommend this manuscript for publication in MDPI Pharmaceutics. The responses are inadequate, nor are clearly explained. Of particular interest is the lack of commentary about heterogeneity of presented nano formulation, and lack of definitive proof of their successful assembly as described. My questions were captured as major comments no.s 2,3,5 and 6 in my first review.

Reviewer 3 Report

In my opinion, the author has not addressed the given suggestion sincerely. I am not convinced by the revision.  

1. , In my opinion, findings should also be incorporated in the revised manuscript in a form of absolute results in abstract section.

2. Authors have to incorporate the formulation-related literature carried by other researchers and highlight the current research gap in the introduction section. It is suggested to link the current investigation to how helpful to fulfill the formulation-related research gap including the merits of current investigation.

3. The author has to highlight/label the different structures in the tissues of the liver, heart, lungs, spleen, and kidney to show no changes observed on those labeled structures of the tissues and observed similar to normal tissues, if any fewer significant changes are observed compared to normal tissues then it should be incorporated in histopathology investigations.